# Recruiting Perovskites to Degrade Toxic Trinitrotoluene

**DOI:** 10.3390/ma14237387

**Published:** 2021-12-02

**Authors:** Yuri A. Mastrikov, Roman Tsyshevsky, Fenggong Wang, Maija M. Kuklja

**Affiliations:** Department of Materials Science and Engineering, University of Maryland, College Park, MD 20742, USA; yuri@umd.edu (Y.A.M.); rtsyshev@umd.edu (R.T.); fenggong.wang@gmail.com (F.W.)

**Keywords:** chemical decomposition mechanisms, catalytic degradation, high energy density materials, high explosives, reaction activation barriers and kinetics

## Abstract

Everybody knows TNT, the most widely used explosive material and a universal measure of the destructiveness of explosions. A long history of use and extensive manufacture of toxic TNT leads to the accumulation of these materials in soil and groundwater, which is a significant concern for environmental safety and sustainability. Reliable and cost-efficient technologies for removing or detoxifying TNT from the environment are lacking. Despite the extreme urgency, this remains an outstanding challenge that often goes unnoticed. We report here that highly controlled energy release from explosive molecules can be accomplished rather easily by preparing TNT–perovskite mixtures with a tailored perovskite surface morphology at ambient conditions. These results offer new insight into understanding the sensitivity of high explosives to detonation initiation and enable many novel applications, such as new concepts in harvesting and converting chemical energy, the design of new, improved energetics with tunable characteristics, the development of powerful fuels and miniaturized detonators, and new ways for eliminating toxins from land and water.

## 1. Introduction

Trinitrotoluene (also known as TNT, C_6_H_2_(NO_2_)_3_CH_3_, trotyl) is among the most famous and perhaps most influential materials in the world. Fundamental mechanisms of TNT decomposition under various conditions, however, have yet to be established. TNT is best known as an explosive and its explosive yield is considered the standard comparative convention for expressing energy, typically used to describe the energy released in an explosion [1]. Originally obtained in 1863 as a yellow dye [2], TNT was realized to have explosive properties only 30 years later [3]. Since the early 20th century through to today, TNT has remained the most common explosive for military, industrial, and mining applications because of its compatibility with other materials, low hygroscopicity, low melting point, low cost, relatively low sensitivity to impact and friction, good thermal stability, high power during an explosion, and moderate toxicity [4,5,6]. Occasionally, TNT is used for the synthesis of other explosives [1,7], cocrystals [8,9], composites [1,10], and even diamonds [11,12,13,14].

Despite a long history and vast experience of the widespread use of TNT, a fundamental understanding of micro-scale mechanisms of explosive chemistry is largely lacking. In particular, factors that govern thermal stability (also referred to as the sensitivity to the initiation of detonation, that is, a measure of how much thermal energy is needed to initiate a chemical decomposition) of energetic materials are not fully understood [15,16,17,18,19]. TNT is very stable and is often used in energetic formulations to form either blends [20,21,22,23,24], coatings [25,26], or polymorphic crystals [8,9,27] in efforts focused on increasing stability (i.e., reducing its sensitivity to external perturbation) of high explosive compositions.

Conventionally, the main challenge in the field of high explosives is to determine new methods or techniques that would deliver energetic materials with low sensitivity to the initiation of detonation (that is, to ensure the explosives explode only upon command and not accidently) and high performance (that is, to have materials with a high energy release reflect this in a large amount of destructive power). Certainly, one desires controllable, safe, and predictable behavior of high energy density materials to make them most useful. In the meantime, there is yet another challenge, which is discussed in the research community less frequently. The challenge is related to environmental protection and the need for eliminating toxic explosives from land and water [18,19,20,21,22,23,24,25,26,27,28,29,30,31]. Worldwide land and water are contaminated by energetic materials from military conflicts and training activities, manufacturing operations, rocket fuels, dumping, and open burning of obsolete munitions [29,32]. When released to the biosphere, energetics are contaminants posing toxic hazards to ecosystems, animals, plants, and humans. For instance, TNT is a long-known mutagen, and its toxicity and its degradation products are extensively documented. In humans, TNT can cause dermatitis, vomiting, toxic hepatitis and liver damage, methemoglobinemia, and aplastic anemia, which affects blood cell production [33,34,35].

Typically, energetic materials are discarded through open burning/open detonation, or in landfills. Although many disposal technologies (including activated carbon adsorption [36,37,38], photocatalysis [39,40,41], oxidation [42,43], biodegradation [44,45,46,47,48,49,50,51], hydrothermal processing [52], and electric discharge [53] etc.) are being explored, no reliable, safe, environmentally sound, and cost-efficient means of disposal are available yet [47,48,54]. Therefore, the development of new ways of eliminating energetic materials and related toxins from land and water remains among the urgent concerns to improve environmental safety and sustainability.

In this work, we discovered the sensitivity of TNT can be dramatically altered on interfaces with perovskites, which trigger the thermal decomposition of TNT with surprisingly low activation energy barriers. To explore this behavior, we conducted quantum-chemical computational modeling of trinitrotoluene (Figure 1a) interacting with a series of perovskites (Figure 1b). Intriguingly, this discovery not only provides new insight into the complexity of the sensitivity of energetic materials to external perturbation, but also reveals fundamentally new opportunities for using perovskites to catalyze a controllable release of energy from explosives and eliminate toxic explosives from land and water.

## 2. Details of Calculations

As a model system to explore the behavior of TNT adsorbed on perovskites, we designed large supercells in which a single TNT molecule is adsorbed on a select surface of a series of TiO_2_-based perovskites. Adsorption and decomposition of TNT on BaTiO_3_ (BTO), SrTiO_3_ (STO), and Ba_0_._5_Sr_0_._5_TiO_3_ (BSTO) were simulated using the PBE exchange-correlation functional [55] as implemented in the computer code VASP 5.4 [56]. The plane wave basis set kinetic cutoff energy was set to 520 eV in all calculations. Atomic cores were substituted with ultra-soft potentials with the PAW approximation [57,58] (Appendix A). In modeling ideal bulk crystals, we used an automatically generated 6 × 6 × 6 *k*-point mesh. The convergence criterion for electronic steps was set to 10^−5^ eV, and the maximum force acting on any atom was set not to exceed 0.02 eV/Å.

Calculations of TNT (Figure 1a) adsorption and decomposition on perovskites were limited to considering the most stable (001) surface [59] only (Figure 1b). To analyze TNT adsorption on STO (Figure 1c), BTO (Figure 1d), and BSTO (Figure 1e), we constructed 4*a*_0_ × 4*a*_0_ surface supercell slabs (Figure 1b). Each symmetric 288 atomic slab contained 7 atomic layers with an upper layer terminated by TiO_2_ and a central layer formed by *A*O atoms (*A* = Sr or Ba). An even number of TiO_2_ planes in the distorted perovskite (001) slabs provides a balanced alternation of the oxygen octahedra tilt, reducing the total stress in the structure [60]. The supercell slabs were separated by a 20 Å vacuum gap in the ***z*** direction to prevent any significant overlap between the electron densities along the direction normal to the surface.

The large supercell slabs, used in our calculations, ensure that a proper lattice relaxation is accounted for in the initial, transition, and final state of each chemical reaction under consideration. In addition, such a large size of the surface supercell allows the Brillouin zone to be sampled by a single *k*-point − Г [61]. For the bare surface of all three perovskites, cell lattice vectors were fully relaxed with no symmetry constraints, and after that, they were kept fixed for all modelled surface reactions. The calculated lattice parameters of STO, BTO, and BSTO crystals are in good agreement with experimental results [62,63] (Appendix A of SI). The pre-exponential factor for reactions with the well-defined transition state can be calculated by applying the transition state theory (TST) using Equation (1) [64]:
(1)$$k_{TST}=A_{TST}e^{-\Delta E/K_{b}T}=\frac{K_{b}T}{h}\frac{q_{vib}^{\#}q_{rot}^{\#}q_{trans}^{\#}}{q_{vib}q_{rot}q_{trans}}e^{-\Delta E/K_{b}T}$$
where *A_TST_* is the pre-exponential factor, *K*_b_ is the Boltzmann constant, *h* is the Planck constant, *T* is the temperature, $q_{vib}^{\#}$, $q_{rot}^{\#}$, and $q_{trans}^{\#}$ are vibrational, rotational, and translational partition functions of the transition state, $q_{vib}$, $q_{rot}$, and $q_{trans}$ are vibrational, rotational, and translational partition functions of the reagent, and Δ*E* is the activation barrier calculated as the energy difference between the total energies of the compound at the transition (*E^#^*) state and the reagent (*E*). In calculating the pre-exponential factor for TNT decomposition on perovskites, we take into consideration vibrational partition functions of the transition state and reactant (Equation (2)) and assume that translational and rotational partition functions of the transition state are close to those of the reactant. Results of our previous studies show that this approach provides reasonable estimates of the Arrhenius parameters for decomposition reactions of energetic materials [19,65]
(2)$$A_{TST}=\frac{K_{b}T}{h}\frac{q_{vib}^{\#}}{q_{vib}}$$

Minimal energy paths in the VASP periodic calculations were obtained with the nudged elastic band method [66]. In modeling chemical reactions, atomic positions were relaxed using conjugate gradient and quasi-Newtonian methods within a force tolerance of 0.05 Å/eV. TNT adsorption energies were refined with the VdW-DF functional [67], to account for weak van der Waals interactions.

## 3. Results and Discussion

### 3.1. TNT Adsorption on Perovskites

Based on energetic considerations, the TNT molecule is strongly adsorbed on the TiO_2_-terminated (001) perovskite surface by forming bonds between oxygen atoms of nitro groups and surface Ti cations (Figure 1c–e). In the most favorable configuration, the aromatic ring is situated parallel to the terminating plane. Energies of TNT adsorption on all three perovskites, BTO, STO, and BSTO, are very close to each other (approximately −43 kcal/mol, Figure 1c–e, Table 1) largely due to a similarity of interactions between TNT and TiO_2_ surface atoms, characteristic for these perovskites. The high binding energies imply that TNT can be interfaced with the perovskites, forming stable structures. At the same time, a single TNT molecule absorbed on the perovskite surface serves as a good model system to explore properties of the interfaces as intermolecular interactions. These interactions are neglected in such a model, since they are relatively weak because they are mostly defined by van der Waals forces while all electronic density is fully localized on the molecules.

### 3.2. Perovskite Surface-Enhanced Decomposition of TNT

Further, we simulated the perovskite-enhanced decomposition of TNT molecules at the interfaces. The TNT thermal decomposition is complex due to co-existing dissociation channels (See SI for more detail). While there is a significant difference in how TNT molecules decompose in the gas phase and solid state, is it accepted that the initiation of chemistry in TNT is defined by the interplay of the C-NO_2_ homolysis (Mechanism 1, Figure 2), cyclization (anthranil formation) [4,15] via *aci*-isomerization (Mechanism 2, Figure 2), and NO loss via nitro-nitrite isomerization (Mechanism 3, Figure 2). Mass-spectroscopy measurements suggest that thermal decomposition of TNT in the condensed phase requires 67.3 kcal/mol and is dominated by the C-NO_2_ homolysis above 770 °C [15]. At a lower temperature, oxidation reactions of the CH_3_ group dominate. Activation barriers reported for TNT decomposition below 500 °C vary from 40.9 to 50.7 kcal/mol [15] and strongly depend on experimental conditions, the quality and history of samples, and heating rates.

In exploring the TNT decomposition on a perovskite surface, we limited our study to the modeling of C-NO_2_ homolysis as the most important reaction. The obtained activation barrier of the C-NO_2_ homolysis in the TNT molecule adsorbed on the BSTO surface (path ***A1****–**A2***, Figure 3) is found to require 66.1 kcal/mol, which is in good agreement with the activation energy observed in TNT condensed-phase measurements, with 67.3 kcal/mol [15], and somewhat higher than the calculated energy for the gas-phase, with 58 kcal/mol [68]. The CONO rearrangement reaction in the TNT molecule on the perovskite surface (path ***A1****–**A3***, Figure 3) requires 69 kcal/mol, which is also higher than the calculated energy for the gas-phase at 54.9 kcal/mol [68]. We note that both the reaction mechanism and the obtained activation barrier appear consistent with the previous knowledge on the decomposition of pure condensed TNT. We also notice that the perovskite surface hardly plays any role in the decomposition, merely serving as a support substrate for the molecule and, thus, mimicking the solid-state phase environment.

Intriguingly, we discovered surface-facilitated chemistry, which proceeds very differently. We will now illustrate and analyze the dramatic effect of the perovskite surface on mechanisms and kinetics of TNT decomposition. A very small perturbation that moves the adsorbed TNT molecule closer to the BSTO surface triggers the C-NO_2_ break (reaction ***A1***–***A4,*** Figure 3) with only 18.5 kcal/mol, which is three to four times lower than the experimentally measured activation barrier of the condensed-phase TNT decomposition (67.3 kcal/mol [15]) and the calculated gas-phase energy (58 kcal/mol [68]). More so, the reaction is exothermic and releases 22.6 kcal/mol of energy (Figure 3, Table 1). Both products of the reaction remain on the surface. Breaking of the second C-NO_2_ bond (***A4***–***A5***, Figure 3) is a more demanding process in comparison with the previous step and requires 30.9 kcal/mol, and the reaction step ***A4***–***A5*** is energetically neutral. A similar reaction in the isolated gas-phase TNT molecule requires 124.5 kcal/mol overall [68], which is again about four times higher than that on the perovskite surface.

The reaction rates of the simulated surface-enhanced decomposition mechanisms are plotted in Figure 4 and clearly demonstrate that all the reactions procced at considerably higher rates than the known reported condensed-phase TNT decomposition [4,15]. Among the probed reactions on the BSTO surface, the primary C-NO_2_ bond break (***A1***–***A4***, Figure 4) is kinetically the most favorable reaction due to the lowest activation energy of 18.5 kcal/mol and relatively high pre-exponential factor of log(*A*, s^−1^) = 12.3. The breaking of the second C-NO_2_ bond (***A4*–*A5***, Figure 4) proceeds at lower rates due to a higher activation barrier (30.9 kcal/mol) and lower pre-exponential factors (log(*A*, s^−1^) = 11.9). Since breaking of the second C-NO_2_ bond (***A4*–*A5***) is an endothermic process, ***A5*** will eventually convert back to ***A1***.

Further, we repeated similar calculations on two relevant perovskite surfaces, BTO and STO. The calculated energies are collected in Table 1 and show that TNT decomposition on BTO and STO requires close barriers, 23.8 and 24.5 kcal/mol, and both energies are slightly higher than that on BSTO (18.5 kcal/mol). The decomposition of TNT on the STO surface has a slightly higher pre-exponential factor log(*A*, s^−1^) = 13.1 than on BTO (log(*A*, s^−1^) = 12.2) and BSTO (log(*A*, s^−1^) = 12.3) as shown in Table 1. Table 1 also indicates that the C-NO_2_ bond cleavage on the STO surface yields a significantly lower reaction energy gain (3.7 kcal/mol) than on BTO (25.1 kcal/mol) and BSTO (22.6 kcal/mol). The observed trends may be explained with the analysis of the transition state structures (Figure 5). The transition states of the TNT C-NO_2_ bond-breaking reaction on the STO and BTO (001) surfaces have similar structures (Figure 5a,b) while the transition state on BSTO (Figure 5c) has a different configuration. For example, in comparison with STO and BTO, the TNT molecule on BSTO has a shorter C-O bond (1.40 Å vs. 1.78 Å) and a longer C-N bond (1.76 Å vs. 1.57 Å, Figure 5a–c). Further, on STO and BTO surfaces, a bond is formed between the surface Ti atom and one of the nitro groups (Figure 5a,b) whereas on the BSTO surface (Figure 5c), two nitro groups bind with surface Ti atoms. Hence, the TNT molecule at the transition state interacts with the BSTO (001) surface more strongly than with STO and BTO surfaces. As a result, TNT decomposition on BSTO requires somewhat lower energy than on STO and BTO.

## 4. Discussion and Conclusions

The decomposition of TNT adsorbed on STO, BTO, and BSTO perovskites is explored by means of quantum-chemical calculations and large periodic supercells. We report here the previously unknown behavior of TNT. We predict that the probed perovskite surfaces readily adsorb TNT and rapidly decompose it with radically low activation barriers. Thus, the C-NO_2_ bond dissociation is triggered by only 18.5–24.5 kcal/mol in energy, which is approximately 3–4 times lower than the ordinary decomposition of TNT samples, which requires ~67.3 kcal/mol [15]. The energy gain from this exothermic reaction (22.6–25.1 kcal/mol) on BTO and BSTO is larger than the activation barrier, which means that the reaction will proceed in a self-supported regimen. Interestingly, our recent study of TNT adsorbed on a TiO_2_ surface confirmed that the presence of titania barely affects TNT decomposition energy barriers [69]. Recent molecular dynamics simulations also advocate for intact adsorption of TNT on pristine TiO_2_ rutile (110) surface [70]. In the presence of water and oxygen on the TiO_2_ rutile (110) surface, however, TNT may convert to trinitrobenzoic acid and trinitrobenzaldehyde [70]. The calculated activation barrier of trinitrobenzoic acid formation is 29 kcal/mol, whereas the formation of trinitrobenzaldehyde requires slightly lower energy (27 kcal/mol). The presence of oxygen vacancy on the TiO_2_ rutile (110) surface was suggested to enhance the adsorption of the molecule but have no effect on the decomposition of TNT [70]. In sharp contrast to that, a series of the titanate perovskites, STO, BTO, and BSTO, in our research, exhibit a significantly stronger effect despite the TNT decomposition proceeding on the chemically similar TiO_2_-terminated surfaces. This implies that TNT strongly interacts not only with Ti and O atoms on the surface to which it is bound, but also with A-metals (Sr and Ba) at the sub-surface layer. This, in turn, hints that the possible variations of the perovskite surface morphology (and especially a wealth of crystallographic defects that perovskite lattice can accommodate) would allow for tailored and precise manipulations of the decomposition chemistry and energy release from highly explosive molecules at ambient conditions.

These results offer new insight into the sensitivity of highly energetic composite materials crucially important for the design of new fuels, oxidizers, and improved energetic formulations with tunable performance and sensitivity parameters. Certainly, experimental validation is required to confirm and refine our predictions. Nevertheless, given our findings are accurate, the obtained conclusions also bring to light fundamentally new opportunities in applications of both perovskites and high explosives that are difficult to overestimate. For example, the highly controlled explosive decomposition chemistry will enable thee design and development of novel, simplified, safe, and cost-efficient architectures of miniaturized detonators and new filter materials and devices for the trapping and destruction of toxic nitro energetic materials and byproducts of their synthesis and decomposition in land fields and ground water. Perhaps, most importantly, it will also elucidate new ways to harvest, convert, and utilize chemical energy stored in high-energy-density materials, potentially transforming the field.

## Figures and Tables

**Figure 1 materials-14-07387-f001:**
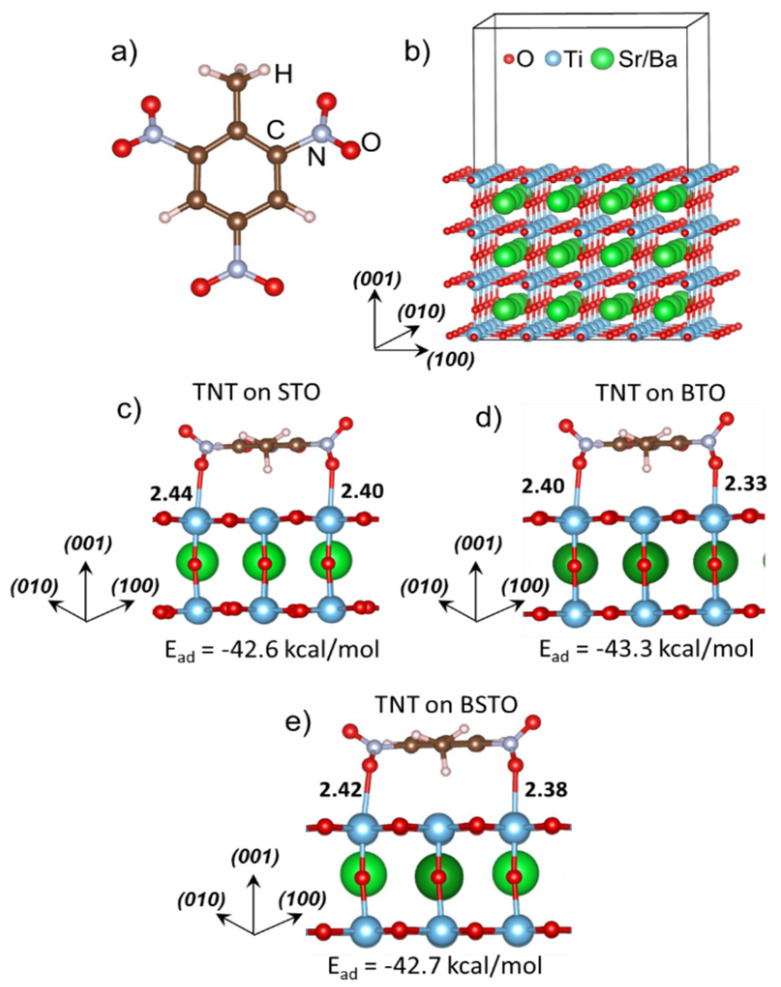
The structures of (**a**) TNT molecule and (**b**) the TiO_2_-terminated 7-plane slab model of *AB*O_3_ (001) perovskite surface. Adsorption of TNT on the TiO_2_-terminated (001) surface of (**c**) STO, (**d**) BTO, and (**e**) BSTO perovskites.

**Figure 2 materials-14-07387-f002:**
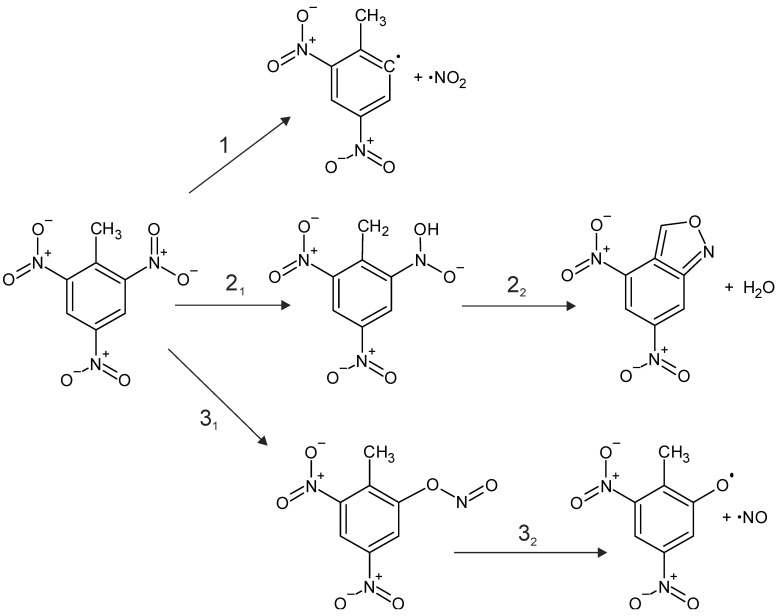
Mechanisms of TNT decomposition.

**Figure 3 materials-14-07387-f003:**
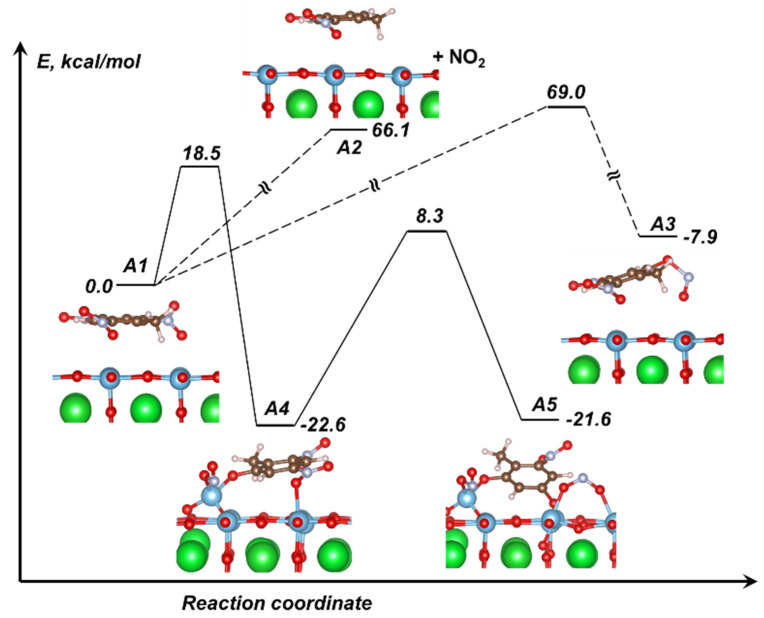
Schematic of TNT decomposition on the BSTO (001) surface via the C-NO_2_ break and CONO isomerization.

**Figure 4 materials-14-07387-f004:**
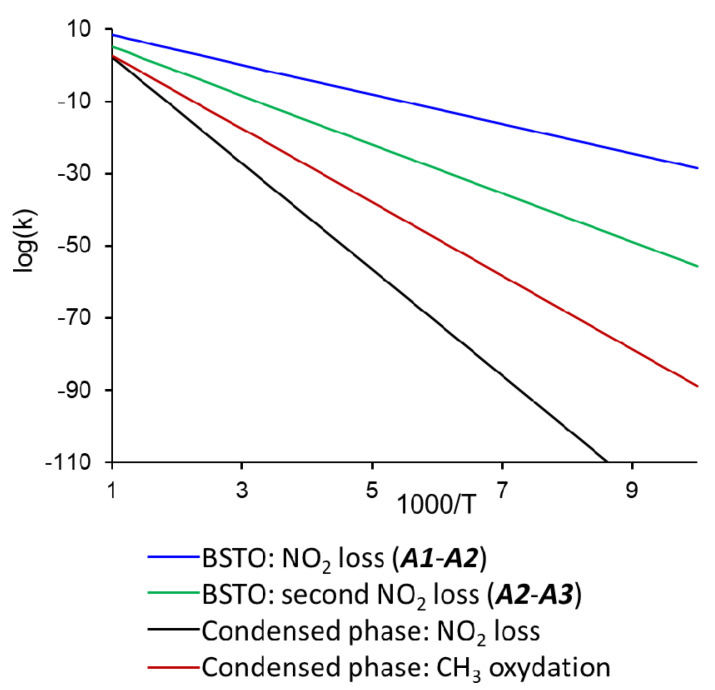
Reactions rates of TNT decomposition on BSTO surface and in the condensed phase. Reaction rates for condensed TNT decomposition were calculated using experimentally measured Arrhenius parameters from [15].

**Figure 5 materials-14-07387-f005:**
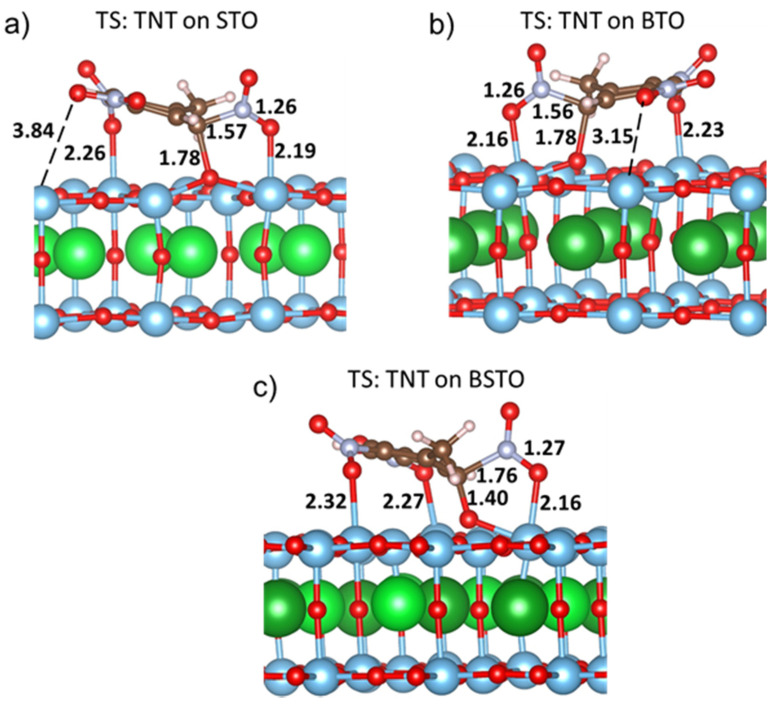
Transition-state structures of the C-NO_2_ bond cleavage in a TNT molecule on (**a**) STO, (**b**) BTO, (**c**) BSTO (001) surfaces (all bond lengths and interatomic distances are in Å).

**Table 1 materials-14-07387-t001:** Obtained energies (E_ad_) of TNT adsorption on the STO, BTO, and BSTO (001) surfaces, activation barriers (ΔE_b_), reaction energies (E_r_), and pre-exponential factors of TNT decomposition on perovskite surfaces via the C-NO_2_ bond cleavage (log(A)).

Material	E_ad,_ kcal/mol	ΔE_b_, kcal/mol	log(A, s^−1^)	E_r_, kcal/mol
SrTiO_3_	−42.6	24.5 ^a^ (22.3) ^b^	13.1	−3.7
Ba_0_._5_Sr_0_._5_TiO_3_	−42.7	18.5 (18.4)	12.3	−22.6
BaTiO_3_	−43.3	23.8 (22.2)	12.2	−25.1

^a^ DFT—calculated energy of C-NO_2_ bond cleavage in gas-phase TNT molecule is 58.0 kcal/mol [68]. ^b^ ZPE—corrected activation barriers are provided in parentheses.

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
