# Peer review of "Recruiting Perovskites to Degrade Toxic Trinitrotoluene"

_materials, 2021, doi:10.3390/ma14237387_

Round 1

Reviewer 1 Report

Please consider the suggestions presented in the revision report attached.

Author Response

To Reviewer 1

  • The conclusions stated by the authors are, considering this theoretical study, somewhat exaggerated, namely in what concerns future application. Before stating all that, this work will led to, it is necessary to prove it experimentally and for more materials than TNT.

Response: We agree with the reviewer that our conclusions obtained purely from theory and simulations have to be experimentally validated and therefore added a statement to reflect this at the end of the conclusion section.  

  • The reference model is different in lines 88-91, please place the references accordingly to the rest of the text.

Response: Corrected.

  • Figure 2 quality should be improved.

Response: An image in Figure 2 is replaced with the image of higher resolution.

  • Uniformize the presentation of the paths of Figure 3 in the text: use Ax-Ay or Ax-Ay and define path Ax-Ay or only Ax-Ay.

Response: The presentation of the reaction paths is uniformized. All pathways are defined in the text.

  • In line 51 the acronym EM is used. What that it stands for? Is it Energetic Materials? The use of the acronym seems unnecessary, but if maintained it needs to be defined.

Response: Acronym is removed.

  • In lines 68-69 it is stated that “maximum force acting on any atom was set not to exceed 0.02 eV/Å”. Why was this value chosen? The same question is valid for “a force tolerance of 0.05 Å /eV” (line 89).

Response: With our vast experience with modeling and using the software, we obtained that these are the optimal convergence parameters of the force tolerance for the geometry relaxation and modeling reaction pathways. A further reduction of these convergence parameters usually tremendously increases computational time but has no meaningful effect on either geometry structures or energies of the systems.  

  • In lines 183-184 the sentence “Thus, the C-NO2 bond dissociation is triggered by only 18.5-24.5 kcal/mol in energy, which is factors lower than the ordinary decomposition of TNT samples (~67.3 kcal/mol [15]).” is incomplete, the numerical value is missing.

Response: The sentence is actually complete, however to remove any confusion, we revised it to state, “Thus, the C-NO2 bond dissociation is triggered by only 18.5-24.5 kcal/mol in energy, which is approximately 3-4 times factors lower than the ordinary decomposition of TNT samples, which requires ~67.3 kcal/mol [15].”

  • Minor typos

Line 26 – “for the synthesis”; Corrected

Lines 29-30 – “also referred as the sensitivity to initiate the detonation, it is a measure of how much thermal energy is needed to initiate a chemical decomposition”; Corrected

Lines 43-44 – “For instance, TNT is a long-known mutagen, its toxicity and its degradation products are extensively documented”; Corrected

Line 53 – “that sensitivity”; Corrected

Line 55 – “series of perovskites”; Corrected

Line 56 – “into the complexity of the sensitivity of energetic”; Corrected

Line 66 – “520 eV”; Corrected

Line 81 – “with experimental results”; Corrected

Line 90 – “TNT adsorption energies”; Corrected

Line 97 – “surface of Ti”; Corrected

Line 99 – “approximately -43 kcal/mol”; Corrected

Lines 101-104 – “At the same time, a single TNT molecule absorbed on the perovskite surface serves as a good model system to explore properties of the interfaces as intermolecular interactions. These interactions are neglected in such a model, since they are relatively weak because they are mostly defined by van der Waals forces while all electronic density is fully localized on molecules”; Corrected

Lines 107-108 – “pre-exponential factors of TNT decomposition on perovskite surfaces via the C-NO2 bond cleavage (log A)”; Corrected 

Line 122 – “Figure 2. Mechanisms of TNT decomposition.” And add a space between the caption and the tex. Corrected 

Line 123 – “study to the modelling”;  Corrected 

Line 127 – “reaction”; Corrected

Line 129 – “kcal/mol [66]. We notice”; Corrected

In Figure 4 replace NO2 for NO2 and CH3 for CH3 in the legend; Corrected

Lines 164-166 – “transition state structures (Fig. 5). The transition states of the TNT C-NO2 bond breaking reaction on the STO and BTO (001) surfaces have alike structures (Fig. 5a and 5b) while the transition state on BSTO (Fig. 5c) has a different configuration.”; Corrected

Line 167 – “(1.40 Å vs 1.78 Å) and longer C-N bond (1.76 Å vs 1.57 Å , Fig. 5 a-c).”; Corrected

Line 187 – “titanium” and not “titania”; Response: This sentence is correct. We refer here to results on TNT decomposition on titanium oxide.

Line 188 – “adsorption”. Corrected

Reviewer 2 Report

The authors in “Recruiting Perovskites to Degrade Toxic Trinitrotoluene” show that TNT adsorbed on STO, BTO, and BSTO perovskites rapidly decompose with low input energy. They use model system to explore the behavior of TNT on various surfaces. It is difficult to understand the scope of this article. Is it about explosives and on how to improve them or on degradation of TNT present in the environment, as suggested by the title and intro?
The title of this article is misleading and this article is not about degradation of toxic TNT but about optimization of the explosive materials, which is of course very important but out of scope of the Special Issue.
This article cannot be accepted for publication in Special Issue of Materials entitled: "New Advances in Heterogeneous Catalysis Materials".

Author Response

To Reviewer 2

The authors in “Recruiting Perovskites to Degrade Toxic Trinitrotoluene” show that TNT adsorbed on STO, BTO, and BSTO perovskites rapidly decompose with low input energy. They use model system to explore the behavior of TNT on various surfaces. It is difficult to understand the scope of this article. Is it about explosives and on how to improve them or on degradation of TNT present in the environment, as suggested by the title and intro?
The title of this article is misleading and this article is not about degradation of toxic TNT but about optimization of the explosive materials, which is of course very important but out of scope of the Special Issue.
This article cannot be accepted for publication in Special Issue of Materials entitled: "New Advances in Heterogeneous Catalysis Materials".

Response: We acknowledge the comment and think that the reviewer answered the question himself/herself. This paper is on both getting better understanding of the behavior of TNT (and more generally, high energy density materials), particularly, their decomposition chemistry, and on ways to efficiently degrade these materials in the environment. While we do not set a goal to improve explosives in this research, we argue here that revealing details of their chemistry will lead to many outcomes, including improved materials and clean environment. Nevertheless, our current research and this paper is focused on fundamental aspects of decomposition chemistry of TNT on perovskite surfaces. We believe that this fits the special issue as one of the main results is that perovskite surfaces facilitate (catalyze) decomposition of TNT with low energy barriers. This is a new result, which potentially will lead to new applications in heterogeneous catalysis and beyond.   

Reviewer 3 Report

Trinitrotoluene (TNT) is among the most famous and perhaps most influential materials in the world. TNT remains the most common explosive for military, industrial, and mining applications and on the other hand, worldwide land and water are contaminated by energetic materials. TNT can cause dermatitis, vomiting, toxic hepatitis and liver damage, etc., and affect blood cell production. Thus, the development of new ways of eliminating TNT related toxins from land and water is urgent. Now Mastrikov et al. used density functional theory to explore the TNT decomposition mechanics and found that highly controlled energy release from high explosive molecules can be rather easily accomplished by preparing TNT-perovskite mixtures with a tailored perovskite surface morphology at ambient conditions. Their results offer a new insight into understanding of sensitivity of TNT and provide a new way for eliminating toxins from land and water. The manuscript is of wide interest and of fine significance. Nevertheless, there are some issues required to be clarified before it could be accepted for publication. The following points are very important to improve the manuscript. They should be addressed and included in the revised manuscript. (1) In Lines 90-91 of Section II, the authors stated that Energies of TNT adsorption energies were refined with the VdW-DF functional, to account for weak van der Waals interactions. This is good for adsorption. However, in the literature, Gholizadeh et al. [i.e., N2O + CO reaction over Si- and Se-doped graphenes: An ab initio DFT study, Appl. Surf. Sci. 2015, 357, 1187-1195.] found that the dispersion corrections are important for the generalized gradient approximation (GGA) while zero point energy (ZPE) contributions can be neglected in most cases except for the reaction barrier calculations. I did not know whether the authors have considered the ZPE corrections to reaction barriers and I recommend the authors use above literature results as evidences for the calculation methods they selected. (2) The values of the pre-exponential factors are listed in Table 1 but the authors did not describe how to obtain these values. (3) The electron transfer in the reaction process is important for us to understand the reaction mechanics of TNT decomposition. Why not to list the electron transfer in the process of TNT decomposition on the BSTO (001) surface via the C-NO2 break and CONO isomerization? Discussion on the electron transfer is also essential to improve the significance of the manuscript. (4) For clarity, please give the equation for the reactions rates of TNT decomposition (may be in Arrhenius form). (5) In Ref. 60, “3” in “double-layered SrTiO3 (001) surfaces” should be corrected to a subscript. (6) In Ref. 68, “2” in “TiO2 (110) surface” should also be corrected to a subscript.

Author Response

To Reviewer 3

  • In Lines 90-91 of Section II, the authors stated that Energies of TNT adsorption energies were refined with the VdW-DF functional, to account for weak van der Waals interactions. This is good for adsorption. However, in the literature, Gholizadeh et al. [i.e., N2O + CO reaction over Si- and Se-doped graphenes: An ab initio DFT study, Appl. Surf. Sci. 2015, 357, 1187-1195.] found that the dispersion corrections are important for the generalized gradient approximation (GGA) while zero point energy (ZPE) contributions can be neglected in most cases except for the reaction barrier calculations. I did not know whether the authors have considered the ZPE corrections to reaction barriers and I recommend the authors use above literature results as evidences for the calculation methods they selected.

Response: We agree with the reviewer. For some systems, an inclusion of ZPE corrections may have a considerable effect on activation barriers, indeed. For example, ZPE-corrected activation barriers of decomposition reactions of energetic materials in the gas phase are typically 5 kcal/mol lower than activations barriers calculated without including ZPE corrections (see Molecules 21 (2016) 236-1-22). However, for reactions of TNT decomposition on perovskites, ZPE corrections have little effect. ZPE-corrected barriers are 0.1-2.2 kcal/mol lower than activations barriers estimated using total energies. Therefore, in this paper, we discuss decomposition mechanisms using values of activation barriers calculated without ZPE corrections. Nonetheless, we included ZPE/corrected barriers in Table 1 for completeness. 

  • The values of the pre-exponential factors are listed in Table 1 but the authors did not describe how to obtain these values.

Response: The methodology has been developed by us some years ago. We agree, however, that it is useful to include a brief statement on how this was done as such calculations are still rare. We added details of calculations of the pre-exponential factors in the Details of Calculations section, including references and main equations.

  • The electron transfer in the reaction process is important for us to understand the reaction mechanics of TNT decomposition. Why not to list the electron transfer in the process of TNT decomposition on the BSTO (001) surface via the C-NO2 break and CONO isomerization? Discussion on the electron transfer is also essential to improve the significance of the manuscript. Response: We agree that the charge transfer represents an intriguing and complex aspect of surface chemistry. An electron transfer can have a considerable impact on mechanisms and activation barriers of surface-facilitated reactions. However, in modeling of TNT decomposition on ideal perovskites’ surfaces, we observed neither any significant charge transfer from the surface to the TNT molecule nor any considerable redistribution of the charge density within the TNT molecule.

The situation might change when surface defects are included in consideration. For example, oxygen vacancy in oxides will usually trap two electrons and some of this electronic density will be transferred to an adsorbed molecule as it was shown for the case of PETN on MgO (see, for example, M.M.Kuklja, Quantum-Chemical Modeling of Energetic Materials: Chemical Reactions Triggered by Defects, Deformations, and Electronic Excitations, Advances in Quantum Chemistry: Energetic Materials - Vol 68, edited by John R. Sabin and Erkki Brändas, Elsevier Inc., 2014, pp. 71-146).

We would like to refer to other relevant papers by our group, in which we studied polar surfaces of perovskites (E. A. Kotomin, R. Merkle, Yu. A. Mastrikov, M. M. Kuklja, J. Maier, The Effect of (La,Sr)MnO3 Cathode Surface Termination on Its Electronic Structure, ECS Transactions, 77 (10) 67-73 (2017), 10.1149/07710.0067ecst ©The Electrochemical Society; Yuri A Mastrikov, Rotraut Merkle, Eugene A Kotomin, Maija M Kuklja, Joachim Maier, Surface termination effects on the oxygen reduction reaction rate at fuel cell cathodes, J. Mater. Chem. A, 2018, 6, 11929-11940, DOI: 10.1039/C8TA02058B) and molecular high explosives (M. Kuklja, R. Tsyshevsky, O. Sharia, Effect of polar surfaces on decomposition of molecular materials, J. Am. Chem. Soc., 2014, 136 (38), pp 13289–13302).

  • For clarity, please give the equation for the reactions rates of TNT decomposition (may be in Arrhenius form).

Response: We added details of calculations of the pre-exponential factors and reaction rate constants in the Details of Calculations section.

  • In Ref. 60, “3” in “double-layered SrTiO3 (001) surfaces” should be corrected to a subscript. Response: Corrected.
  • In Ref. 68, “2” in “TiO2 (110) surface” should also be corrected to a subscript.

Response: Corrected.

Round 2

Reviewer 2 Report

I continue to find the conclusions and future applications exaggerated and not well justified. This is well-performed theoretical work; however, is not providing information on how the proposed system will be applied for eliminating TNT from the land and water.

I find it of interest to the public of Materials and well performed and written. I do not find it suitable for the special issue for which it is proposed. If this is to be included in the proposed special issue the future applications, especially removal of TNT form the land and water, should be deepen.

Author Response

To Reviewer 2

I continue to find the conclusions and future applications exaggerated and not well justified. This is well-performed theoretical work; however, is not providing information on how the proposed system will be applied for eliminating TNT from the land and water.

I find it of interest to the public of Materials and well performed and written. I do not find it suitable for the special issue for which it is proposed. If this is to be included in the proposed special issue the future applications, especially removal of TNT form the land and water, should be deepen.

Response: We appreciate the reviewer’s opinion that we presented a well-performed theoretical work. This paper is focused on fundamental science, specifically, we aimed at getting better understanding of the behavior of TNT (and more generally, high energy density materials and their decomposition chemistry). While we do not set a goal to offer or develop new applications of high explosive materials or improve explosives in this research, we argue here that revealing details of their chemistry will lead to many outcomes, including improved materials and clean environment. Nevertheless, our current research and this paper are focused on fundamental aspects of decomposition chemistry of TNT on perovskite surfaces. We believe that this fits the special issue as one of the main results is that perovskite surfaces facilitate (catalyze) decomposition of TNT with low energy barriers. This is a new result, which potentially will lead to new applications in heterogeneous catalysis and on ways to efficiently degrade these materials in the environment. We believe that the development of a technology to apply our findings is well beyond the scope of our paper. Hence, we only point to the potential applications and outcomes to highlight importance of our research conclusions.

We found the second comment of the reviewer somewhat contradictory as he/she says that the paper is well-written and is of interest to the readership of Materials but does not fit within the special issue. We think that the special issue fits with the general scope of Materials and as such, our paper also fits here.

Reviewer 3 Report

The authors have addressed all my questions and now the revised manuscript has been improved adequately for publication.

Author Response

Noted. Some typos are corrected.